# Current Knowledge of the Entomopathogenic Fungal Species *Metarhizium flavoviride* Sensu Lato and Its Potential in Sustainable Pest Control

**DOI:** 10.3390/insects10110385

**Published:** 2019-11-02

**Authors:** Franciska Tóthné Bogdányi, Renáta Petrikovszki, Adalbert Balog, Barna Putnoky-Csicsó, Anita Gódor, János Bálint, Ferenc Tóth

**Affiliations:** 1FKF Nonprofit Zrt., Alföldi str. 7, 1081 Budapest, Hungary; t.bogdanyi.franciska@gmail.com; 2Plant Protection Institute, Faculty of Agricultural and Environmental Sciences, Szent István University, Páter Károly srt. 1, 2100 Gödöllő, Hungary; petrencsi@gmail.com (R.P.); agodor@gmail.com (A.G.); 3Department of Horticulture, Faculty of Technical and Human Sciences, Sapientia Hungarian University of Transylvania, Allea Sighișoarei 1C, 540485 Targu Mures/Corunca, Romania; adalbert.balog@ms.sapientia.ro (A.B.); csicsobarni@gmail.com (B.P.-C.)

**Keywords:** fermentation, formulation, field crops, greenhouse crops, sustainable management, pest control

## Abstract

Fungal entomopathogens are gaining increasing attention as alternatives to chemical control of arthropod pests, and the literature on their use under different conditions and against different species keeps expanding. Our review compiles information regarding the entomopathogenic fungal species *Metarhizium flavoviride* (Gams and Rozsypal 1956) (Hypocreales: Clavicipitaceae) and gives account of the natural occurrences and target arthropods that can be controlled using *M. flavoviride*. Taxonomic problems around *M. flavoviride* species sensu lato are explained. Bioassays, laboratory and field studies examining the effect of fermentation, culture regimes and formulation are compiled along with studies on the effect of the fungus on target and non-target organisms and presenting the effect of management practices on the use of the fungus. Altogether, we provide information to help conducting basic studies, and by pointing out relatively uncharted territories, help to set new research areas.

## 1. Introduction

As an alternative to chemicals, the use of mycoinsecticides is considered an ecologically friendly method in the control of arthropod pests. More and more fungal strains and isolates are on their way to becoming commercial products available for the market; their use in sustainable pest control is under expansion. Members of the *Metarhizium* genus (Sorokin) seem to have the potential to become successful entomopathogenic agents. Among them, *Metarhizium anisopliae* (Metschnikoff) Sorokin (Hypocreales: Clavicipitaceae) stands out both in the volume of related scientific research, including review articles, and in the use in organic pest control [1]. Some other members of the genus have also been frequently cited. To mention some of the other entomopathogens that have been researched, *M. brunneum* (Petch) was found as a promising candidate against Coleopteran pests (wireworms, larvae of Elateridae) in potatoes and other vegetables [2,3] and *M. brunneum* and *M. robertsii* (J.F. Bischoff, Rehner and Humber) are also on their way to become commercial products against the soil-dwelling life stages of Coleopteran and Dipteran pests (cabbage fly, *Delia radicum*, Linnaeus and the crane fly, *Tipula paludosa*, Meigen) in various crops [4,5,6]. In 2017, a compilation [7] suggested that although its successful application needs further studies and trials, *Metarhizium rileyi* ((Farlow) Kepler, S.A. Rehner and Humber) may also have strong potential to become a commercial product as it was found effective against Lepidopteran pests. Our targeted taxon, *M. flavoviride* has a record of being available as a registered product against scarab larvae in Australia [8,9,10,11], but there is no consensus on the taxonomic status of the active microbial agent used. The first two sources citing Copping (2004) refer to the agent as *M. flavoviride* var. *flavoviride*, and the latter two name the agent *M. flavoviride.* Indeed, the literature available on *M. flavoviride* sensu lato raises questions of taxonomic nature, and while there is an array of tests and investigations on, *Metarhizium flavoviride* sensu lato, the conditions of its use remain relatively unmapped, and its potential has not yet been fully studied and discovered. Indeed, *M. flavoviride* is one of the most rarely investigated and documented fungal entomopathogens [12,13,14].

Therefore, the present paper aims to provide a guide to help readers seeking potential agents in sustainable pest control navigate within and understand the state of present-day knowledge of *M. flavoviride* isolates and strains. We collected and sorted scientific information on *M. flavoviride* sensu lato with the purpose to guide entomologists within the selected literature on this fungus used under different conditions and against different pests. The most important host species and circumstances of use regarding *M. flavoviride* sensu lato are also presented.

## 2. Literatures on the Genus *Metarhizium*


We explored scientific papers from over 70 websites. The most frequent sources of information were online research collections including SpringerLink, ScienceDirect, Taylor and Francis Online, Wiley Online Library, ResearchGate, PubMed, Scientific Electronic Library Online, BioOne Complete, Web of Science and also online journals and databases [15,16,17,18,19,20,21,22,23,24]. Working through the various sections of our manuscript, we selected simple keywords to identify scientific papers to match the main topic of the section. While “*Metarhizium flavoviride*” was always an identifier, other keywords varied according to topic. We arranged the information in a sequence that may reflect the approach of a researcher or extension agent for whom *M. flavoviride* is a relatively new area: going from basic towards more specialized information regarding the potential and actual application. Our review starts with the earliest accounts of the species including its natural occurrence and the range of natural hosts. This is followed by studies on the impact of formulation types, culture media on the use of the various strains and isolates on bioassays, and studies carried out in greenhouses, pot experiments or in fields; regarding the effect of management practices are compiled. Altogether more than 110 articles including reviews, experimental and observational studies were compiled.

## 3. A Short Introduction to the Genus *Metarhizium*

Currently the genus *Metarhizium* is positioned within the Kingdom of Fungi (Mycota), Division Ascomycota, Subdivision Pezizomycotina, Class Sordariomycetes, Subclass Hypocreomycetidae, Order Hypocreales, Family Clavicipitaceae [19].

Members of the genus are frequently isolated worldwide under all types of climate conditions, from the artic to tropical regions, from soil samples of various soil types, and found on a multitude of arthropod taxa [25,26,27,28]. One possible explanation for the ubiquity of the genus is their small and hydrophobic conidia that can be easily transported by the movement of the wind (8). The greenish conidial growth is one of the main general morphological features of the genus [27], hence the initial name of the disease its members cause—‘green muscardine’ [29]. The conidiophores have branch formations that may vary from species to species, and the phialides that may have various shapes may also be single or multiplied [12]. Their facultative saprophytic lifestyle allows them to attack arthropods and grow as a parasite on their bodies. In the absence of a suitable host, members of this genus, due to their labile metabolism can live freely within the rhizosphere of plants or survive on non-living particles within the soil [9,12].

## 4. Questions of Taxonomy and Identification Issues

We understand that the taxonomy of fungal entomopathogens evolved from relying on morphological traits to using molecular traits [30], but as science advances, so has to progress the concept of species as well [31].

The genus *Metarhizium* is a genetically diverse taxon, and the color of the colony, the dimension and measurements of the conidia of different species are not reliable identification factors [32,33]. To some success, simple identification measures have been used with some success: an earlier study suggested distinguishing between *M. anisopliae* and *M. flavoviride* based on the difference in the homogeneity of blastophores, as the blastophores of *M. flavoviride* isolates were found more homogenous when compared to those of *M. anisopliae* [34]. Further, a recent attempt to simplify the identification of species or species complexes by evaluating certain isolates on their heat-tolerance and cold activity has been successful in the case of *M. anisopliae* var*. anisopliae, M. anisopliae* var. acridum and two isolates of the *M. flavoviride* species complex [35].

Nevertheless, diversity within an isolate makes identification difficult: morphological features of the same isolate is influenced by the age of culture, the substrate the culture is kept on, and on certain environmental conditions like temperature [29,36].

Around the turn of the century, as biochemical and molecular studies have become more and more used and accepted, papers have begun to point out that some, if not almost all of the isolates listed on the basis of morphological features as *M. flavoviride* or *M. anisopliae,* especially those that used Acridiid target species are actually *M. acridum* [36,37,38].

Research papers since the first reassessment of the genus based on molecular phylogenetics [29] have come to the agreement that the most accurate delimitation is based on molecular analysis, and the designation of species is best based on definitive genetic markers such as the random amplified polymorphic DNA (*RAPD*) markers and the nucleotide sequences of the internal transcribed spacer (ITS) region [14,25,26,32,39].

ITS-sequence phylogenetic analysis and multigene phylogenetic methods have laid a solid foundation for one of the latest descriptions of the *M. anisopliae* and *M. flavoviride* complexes [27,40]. At the same time, the definition of species is an inconclusive matter of taxonomic debates. When the definition of evolutionary units is unreliable, the relationship between two specimens or two taxa also becomes unreliable. This makes the concept of species a hypothesis, and therefore misleading, because we hardly ever have the chance to test whether two specimens belong to the same species [41]. This is especially the case in microbiology, where the subject of studies are isolates, not species, which questions the relevance of the species definition of the Linnaean taxonomy [31].

The concept of a nominal taxon, and therefore, nominal species indicates that when a taxon has a name attached to it, and references are made to it by that name regardless of that name being accepted or not, that name defines the taxon in question [42]. A nominal species is therefore a type of taxon below genus level that is defined by its name.

Some argue that the traditional definition of species is not applicable to members of the genus [27,29], and morphospecies comprise a collection of morphologically identical, but separate species [27,43]. These “cryptic” species may have different ecological roles, physiological properties and environmental preferences [43,44], but it is also possible that some of these cryptic species within a morphospecies differ from the rest of the species of the genus and form a group by having the same ecological and environmental characteristics [43]. For the purposes of our study we applied morphospecies as described in [43], because we wanted to map the collection of knowledge about a biological control agent.

## 5. The Species Spectrum of the Genus *Metarhizium*

Based on comprehensive species lists of the genus up to 2014, a phylogenetic survey of isolates conducted in 2017, and on recent reports describing new species, we provide a compilation that contains members (species and variations) of the *Metarhizium* genus (Table 1).

## 6. A Short Literature Overview of the *M. flavoviride* Species Complex

Following the initial detection of the genus *Metarhizium*, Gams and Rozsypal (at times misspelled by authors as Rozypal, Roszypal or Rozsypa) described a new species by the name *M. flavoviride* in 1973 [49]. In 1976 *M. anisopliae*, its two varieties and *M. flavoviride* were recognized as species of the genus. The distinction between the two species was based only on the shape of conidia and the color of the colonies [46].

The spectrum of the *M. flavoviride* species complex was gradually built up. In 1986, *Mf* var*. flavoviride* and var. *minus* was acknowledged by Rombach et al. [29,39]. In 2000, based on molecular analysis of isolates, the characteristics of *M. flavoviride* var*. flavoviride* were found only in the original strain collected and conserved by the original authors of the species, and *Mf.* var*. novozealandicum* and *Mf* var*. pemphigum* were introduced as new variations of the main species [29,39].

In 2005, there were still only three species (*M. anisopliae, M. album* and *M. flavoviride*) recognized within the genus, and suggestions had been made to unify them as *M. anisopliae* [26]. However, the following years seem to have witnessed an opposite trend: it appears that the number of separate species is rising. The *frigidum* variation of the Ma-complex was found to have a closer relationship with the *M. flavoviride* complex [29], and in 2006 was designated as a distinct species, with the name *M. frigidum* [40]. A new species, with a name that refers to the geographical origin of the first isolate was described in 2014 as *M. koreanum* [12]. Advances in molecular tests have led to former variations being elevated to species level in 2014 as *M. minus, M. novozealandicum* and *M. pemphigi* [12]. The location of *M. novozealandicum* within the genus has recently been challenged, and this species has been transferred outside the *Metarhizium flavoviride* species complex MFSC [14]. An isolate that was found a member of the MFSC in 2000 as Mf Type E [29] was assigned species level as *M. brasiliense* as well [12]. In 2016, a new species, *M. blattodea* was described from Brazil [47]. Recent studies In Japan have expanded the limits of the Mf species complex by adding two new species described as *M. bibionidarum*, which was isolated from Japanese and French soils alike, and was found to have a close relationship with *M. pemphigi*, and *M. purpureogenum*, a remote species with unique conidial shape within the complex, where the name refers to the distinct pigment production of the species [14].

## 7. Natural Occurrences and Natural Hosts of *M. flavoviride*

*Metarhizium flavoviride* was first observed in Europe in the late 1950s on various life stages (larvae and pupae) of two curculionid beetles: *Ceutorrhynchus macula-alba* Herbst (the poppy capsule weevil) and *C. albovittatus* Germar [49]. According to current classification, the two Coleopteran beetles belong, along with *N. smyrnensis*, to the genus *Neoglocianus* [16]. In 1969 the same fungus was found in agricultural soils in Northern Europe in Germany and the Netherlands [49].

The species was first isolated in another continent, Australia, from a native Orthopteran, *Austracris guttulosa* Walker in 1979 [36]. The first African account of *M. flavoviride* dates from the early 1990s when the presence of the fungus was detected on heavily infected, but still alive *Zonocerus variegatus* Linnaeus and *Hieroglyphus daganensis* Krauss, two local Orthopteran pests in Southern Benin, Africa [50,51]. An intensive study of 350 cadavers of *Locusta migratoria migratorioides* Reiche and Fairmaire (Orthoptera: Acrididae) in the south western part of Madagascar resulted in finding *M. flavoviride* in two of the specimens [34]. The second time a *M. flavoviride* isolate was recorded in Australia was in 1997 [36], and a strain of the species was first detected on an Orthopteran host in the Revillagigedo Islands, México [52]. Natural occurrences of epizootics in Africa were recorded in the late 1990s affecting two orthopterans, *Ornithacris cavroisi* Finot in Niger and *Diabolocatantops axillaris* Burmeister in Chad [51]. In 1997, Mietkiewski et al. examined the presence of fungal entomopathogens in cultivated fields and recorded a rare occurrence of *M. flavoviride* within barley fields [53].

Investigations for the presence of *M. flavoviride* and the frequency of fungal infections were carried out for three consecutive years in Northern Benin, Africa in the 1990s. The occurrence of the fungus on sampled areas was low (1.6% to 2.6%). The dominant hosts of the fungus were orthopterans, mostly those living within the soil or in the surface of the soil (*Acrotylus blondeli* Saussure, *C. senegalensis*, now: *Oedaleus senegalensis* Krauss, *Pyrgomorpha cognata* Krauss and member of the genus *Stenohippus* Uvarov) and those living on trees (*Cryptocatantops haemorrhoidalis* Krauss, *Catantops stramineus* Walker, *Diabolocatantops axillaris* Thunberg and *Harpezocatantops stylifer* Krauss). The rate of infected hosts was less than 3.2%. The study concluded in finding no significant differences between the frequency of infections between years or location. A unique observation of this study is the reddish hue of the conidial mass of *M. flavoviride* on the surface of arthropod cadavers before sporulation [54].

In a 2005 study conducted on viable microorganisms found in a lignite excavation site in Slovakia [55] the presence of *M. flavoviride* was rendered likely by morphological and genetic sequence analysis. Meyling and Eilenberg (2006) were looking for entomopathogenic fungi on different agricultural habitats and while *M. flavoviride* was found to be the third most frequent species in the field in both years, it was hardly collected from adjacent hedgerows [56].

A *M. flavoviride* var. *flavoviride* isolation was reported as a novel isolate from the Philippines, Southeast Asia in 2011. A diseased Lepidopteran instar of *Helicoverpa armigera* Hübner (Lepidoptera: Noctuidae) was detected with clear signs of fungal infection. Morphological characterization designated the isolates grown on the cadaver as the genus *Metarhizium* and molecular analysis (DNA sequencing) confirmed the species status: it was a *M. flavoviride* var. *flavoviride* isolate [57]. In the same year, a thorough investigation involved root sampling of plants of different taxa in the USA, North America. Using an adjusted version of the original *Galleria* bait method by Zimmerman [58], the fungal composition of the root zone of strawberry (Rosaceae), blueberry (Ericaceae), grape (Vitaceae) and various pines (Pinaceae) yielded four *Metarhizium* species. Molecular phylogenetic identification revealed the presence of *M. flavoviride* var. *pemphigi*, a member of the *M. flavoviride* species complex, within the root zone of strawberries and pines [59].

The *Galleria* bait method applied to soil samples collected in Korea, and the subsequent DNA extraction and sequence analysis resulted in confirming the presence of *M. flavoviride* var. *pemphigum* (sic) [60]. When soil samples were taken from a field and its hedgerow in Denmark and were evaluated for the distribution and abundance of *Metarhizium* species by using *Tenebrio molitor* Linnaeus (Coleoptera: Tenebrionidae) larvae as bait, the presence of *M. flavoviride* was proven both in the field and in the hedgerow both by morphological and molecular tests [61]. In 2015, the soil bait method was successful in finding isolates for a study that combined root and soil sampling of two crop fields (winter wheat and winter oilseed rape) and a grass pasture that had been set aside for two decades in Denmark.

In 2015, genetic characterization studies performed on fungal isolates obtained from fungus infected larvae of the coleopteran *Amphimallon solstitiale* Linnaeus (Coleoptera: *Scarabaeidae*) collected from roots of various plants in North Eastern Turkey revealed that the hosts were infected by *M. flavoviride* [62].

The modified *Galleria* bait method, using *Tenebrio molitor* larvae resulted in *M. flavoviride* being the predominant species in the investigated areas. Morphological markers such as conidial colour and dimensions suggested *M. flavoviride*, which was validated by PCR amplification and sequencing analysis as well. It turned out that over 89% of the isolates belonged to *M. flavoviride*, and an amplified fragment length polymorphism (AFLP) analysis revealed high diversity within the species [13].

The occurrence of *M. flavoviride* in a hydrothermal cave was first reported in 2017. The presence of the species was confirmed by morphological, trophic and physiological observations, and the conventional tests were accompanied and confirmed by analysing molecular markers as well [63]. A 2018 study based on *Tenebrio molitor* larvae as bait investigating soil samples collected in Korea found fungal isolates belonging to 12 genera and 29 species, *M. flavoviride* being one of them [8].

The natural occurrence of *M. flavoviride* sensu lato has been documented from a wide range of natural environments, but this does not imply that the taxon can be isolated from all soil samples with the same success. In their argument supported by contemporary literature, [60] presents their finding that in undisturbed, permanent cultures such as riparian areas, natural vegetation is more likely to supply entomopathogenic fungi (and *M. flavoviride* in particular) in higher percentages than agricultural areas. Furthermore, no clear connection was proven between the genetic composition of *Metarhizium* isolates found in agricultural fields and the type of the crop [64]. Furthermore, it continues to be the subject of further examination whether yields are similar between lands under permaculture or conventional agricultural management.

As researchers continue to seek biological alternatives to chemical management protocols, more and more type of habitats and even microhabitats are expected to be investigated for the presence of fungal entomopathogens and of *M. flavoviride*. This expansion of knowledge about the ecology of the taxon will contribute to its more to its more frequent, and possibly more successful use in biological control.

## 8. Effect of Conditions during Fermentation, Cultivation, Culture Regimes on the Performance of *M. flavoviride*

Conditions of fermentation and the composition of the culture medium may significantly influence the efficacy of *M. flavoviride* strains. The first studies tested the effect of medium content on conidial production and concentrations including mortality of target pests, and secondary sporulation. According to the results, the cumulative mortality of *Zonocerus variegatus* L. (Orthoptera: Pyrgomorphidae) increased and the fungal pathogen was highly effective even in samples taken 8 days after spraying with *M. flavoviride* as an oil formulated product [50,65].

Other studies tested the effects the length of storage, and temperature had on *M. flavoviride* [66]. When storage involved the addition of powder and/or fatty acids, germination percentages were positively influenced by carbon:nitrogen [C:N] ratio, age of culture (time passed after inoculation), air moisture and storage method (dry powder or in oil) [67,68,69].

Conidial production, morphological features, fungal pathogenicity, conidial growth, production of conidia and blastospores under growing media also increased when N sources in the medium increased and the C:N ratio also had a positive effect on the pathogenicity of *M. flavoviride* [70,71]. It appears that although an array of potential components of fermentation and culture regimes have been investigated over time, the time frame in which these studies were done is a relatively narrow one, and the field has been somewhat neglected in the past two decades. Therefore, the impact of fermentation on the adaptability and pathogenicity of *M. flavoviride* sensu lato still awaits more detailed tests.

## 9. Effect of Formulation on the Performance of *M. flavoviride*

To successfully shift a fungal candidate from the laboratory to the greenhouse and from the relatively safe and controlled environment of protected production to circumstances found in arable fields, mainly depends on the way the potential pest management agent itself is protected. This protection might be supplied through the appropriate formulation of the product [72].

Alternative formulations (mineral, natural oil and/or water water-based formulations) were tested under various conditions including different wavelength of solar radiation, age of cultures, addition of oils and/or sunscreens [66,67,73,74,75].

As a general observation, a definite positive effect of natural oil formulation was recorded. The effects of formulation may have been expressed in different characteristics of the fungus including conidial growth, viability and potential to cause mortality to target organisms, durability or resistance to certain environmental factors. One may also notice that while this area was intensively tested in the 1990s, less research has been conducted since that time. One inevitable challenge formulation faces when trying to enhance the efficiency of the fungal entomopathogen is the presence of ultra-violet light among unprotected field conditions that have a significant negative effect on *M. flavoviride* germination [76].

Caged field trials testing the efficacy of *M. flavoviride* against target organisms such as *Ostrinia nubilalis, Sesamia cretica* and *Chilo agamemnon* revealed that while nano-formulation enhances the virulence of fungus spores against corn pests, it also results in environmental factors including sunlight, C:N ratio and ultra violet light being less detrimental to fungal spores [77,78]. Although there have been many studies in this area, further, formulation-related research areas may open to address other environmental factors the fungal product may face during its use in protected growing conditions or out in the field.

## 10. Laboratory Studies and Caged Field Trials Testing the Efficacy of *M. flavoviride*

Several methods have been used to test the efficacy of *M. flavoviride* in bioassays against target organisms under laboratory and semi-open field conditions (using cages as meta-environments). Conidial suspension by spinning disc applicator was used against Homoptera: Delphacidae [79], fungal inoculum applied directly to the body was used against Orthoptera: Acrididae [80] while immersion to conidial suspension and inoculated near the mouthpart of the body were tested on Coleoptera: Curculionidae and Orthoptera: Acrididae and Phalacridae [81,82]. A significantly positive effect was detected when spores were applied as oil or water-based suspensions, but conidial dosage and relative humidity increased the mortality of target pests in all cases. Caged field studies revealed that 11% of tested bees (Hymenoptera: Apidae) became infected, while the IMI 330189 formulated *M. flavoviride* spores showed moderate virulence to termites and none at all to several species of beetles, to weevils, coreid bugs, ants and cockroaches [83]. Several studies tested the effect of *M. flavoviride* on Orthoptera: Acrididae when the targets were sprayed with blastophore suspension [34], inoculated with conidial suspension [84], fed by baits inoculated with fungus [85], topically administered with the fungus [86,87,88] and in some studies, the consumption of leaves treated with fungal suspension by target species was observed [70,85,89]. Some of these studies recorded a remarkable adverse effect of the fungus on the food intake of infected nymphs and adults of Orthoptera: Acrididae [85,89]. Later studies also detected a similar effect on other target pests including Lepidoptera: Noctuidae, Pyralidae and Crambidae; when nano-formulated fungus spores of *M. flavoviride* were offered ad libitum [77,78] and against Coleoptera: Tenebrionidae; and when the administration of different conidial concentrations of *M. flavoviride* reduced the infection of wheat by *Fusarium culmorum* [90]. The importance of bioassays is also reflected in the number and stable frequency of bioassay studies over the decades (Table 2). As pest control faces new challenges with the introduction and spread of arthropod species into regions where they were previously unknown, *M. flavoviride* may offer a solution and screening for its efficacy against invasive arthropods is advisable.

## 11. Efficacy of *M. flavoviride* under Greenhouse and Open Field Conditions

Greenhouse and open field studies take the promising candidate, a fungal strain or isolate a step closer to the conditions of actual sustainable crop protection. The outcome of studies testing the pathogenicity of *M. flavoviride* in a near-realistic environment may redefine the limitations of a fungus-based product. The first and effective open field trials using *M. flavoviride* were conducted by Lomer et al. [50,51], when the effect of the fungus on Orthoptera: Pyrgomorphidae was tested in a mixed-vegetable field. Later studies also demonstrated that *M. flavoviride* can be an effective biological control agent against several pests including Orthoptera: Acrididae and Gryllidae in vegetable crops [36,51,91] and Lepidoptera: Noctuidae, Pyralidae and Crambidae in corn [77,78] and Lepidoptera: Gelechiidae in potato under both field and greenhouse conditions [77]. Some of these studies used nano-formulated spores, a formulation type that proved to be one of the most effective in the case of *M. flavoviride*. Although the effectiveness of *M. flavoviride* have been tested and demonstrated under field conditions, so far only a few studies (incl. [36]) added extra method variables (spray bands, aerial and mounted spray) that were proven effective against Orthopteran species.

## 12. Studies on the Compatibility of Management Types and Agricultural Substances on *M. flavoviride*

A fungus-based product in actual use has to overcome challenges posed by variable in the agricultural environment including the use of chemical and organic pesticides and fertilizers, both chemical and organic. The amount and type of tillage may also have an influence on the performance of an otherwise promising strain or isolate.

The number of studies testing the antagonistic effect of certain agricultural compounds on *M. flavoviride* is surprisingly low. It appears that although *M. flavoviride* is available as a ready-to-use biopesticide, there has not been any detailed study to examine the influence of soil type, tillage, or lack of tillage, mulching, irrigation method, or the co-presence of chemical or organic substances, on the performance of *M. flavoviride*.

In laboratories, the growth of *M. flavoviride* was restricted by fungicides in two studies: the first had benomyl [53], the second had carbendazim, a mixture of trifloxystrobin and tebuconazole [92]. The deleterious effect of pesticides on fungal efficacy was also confirmed in the laboratory, when fungal entomopathogens (including *Metarhizium* sp.) were isolated with the Galleria-method from barley fields that were receiving various pesticides (two fungicides, two insecticides, an herbicide) for an extended period of time [53]. The authors argue that the effect of pesticides on fungal performance may vary greatly as there are a multitude of biotic and abiotic factors in the environment to influence the viability, abundance and efficacy of entomopathogenic fungi.

The effect of agricultural management on the performance of *M. flavoviride* awaits further studies, but meaningful differences were found between the species richness of fungal entomopathogens in organic and conventional fields [93]: when conventional and organic fertilizers (N, P, K; pig slurry, green manure) were tested under field conditions, the positive effect of the organic amendments was proven even when they were involved in a small portion of the total amount of nutrients supplied; thereby confirming that fertilizers used in integrated management may have some beneficial effect on *M. flavoviride* [94,95].

## 13. Conclusions

Our aims were to revise the scientific knowledge of *M. flavoviride* and to provide a comprehensive review on its production, formulation, use and effectiveness against arthropod pests (Table 2). It appears that not every member of the *Metarhizium* genus received equal shares of scientific attention. We conclude that more studies are needed to investigate the rhizosphere and document the behavior of *M. flavoviride* and give account of its potential endophytic nature. We are also yet to discover the extent of the plant supportive and soil enhancing effects of *M. flavoviride*. We suggest that more effort put into exploring a variety of microenvironments in search of *M. flavoviride*, and similarly, more effort put into understanding the circumstances under which the species is able to perform well. To understand the limits of fungal performance, we are still in need of defining and tailoring fermentation and formulation protocols to specific application conditions. An array of environmental factors needs exploration in relation to the performance of *M. flavoviride*. It can also be suggested, that more studies are needed with an increased range and type of test variable to help *M. flavoviride* strains and isolates become safe and reliable biocontrol agents and commercial products. We envisage studies exploring the compatibility of *M. flavoviride* as a fungal entomopathogen with elements of agricultural management, tillage, agrochemicals, and herbal products. Finally, a series of intricate tests are advised to investigate the effects of *M. flavoviride* as an effective control agent in sustainable management and its effects on non-target organisms, and also there is a need to explore the potential risk the species may pose to human health.

## Figures and Tables

**Table 1 insects-10-00385-t001:** Alphabetical species list of the *Metarhizium* genus. The first column presents the recent name of the taxon and references. The second column contains the most recent former name (and a reference, where applicable). X = Species of the *M. flavoviride* species complex (MFSC).

Name of Species	Former Name	MFSC
*Metarhizium acridum* (Driver and Milner) J.F. Bisch., Rehner and Humber stat. nov. [25,27]	*Metarhizium anisopliae* var. *acridum* Driver and Milner [29]	
*Metarhizium album* [29]		
*Metarhizium alvesii* Lopes, Faria, Montalva and Humber sp. nov. [45]		
*Metarhizium anisopliae* (Metschn.) Sorokīn [12,25,27]		
*Metarhizium anisopliae* var. *anisopliae* [29]		
*Metarhizium anisopliae* var. *majus* [29] syn. *Metarhizium anisopliae* var. *major* (J.R. Johnst.) M.C. Tulloch [46]		
*Metarhizium atrovirens* (Kobayasi and Shimizu) Kepler, S.A. Rehner and Humber, comb. nov. [12]		
*Metarhizium bibionidarum* O. Nishi, H. Sato, sp. nov. [14] x		X
*Metarhizium blattodeae* Montalva, Humber, Collier and Luz, sp. nov. [47] x		X
*Metarhizium brasiliense* Kepler, S.A. Rehner and Humber, sp. nov. [12]	*Metarhizium flavoviride* Type E [29] x	
*Metarhizium brittlebankisoides* (Zuo Y. Liu, Z.Q. Liang, Whalley, Y.J. Yao and A.Y. Liu) Kepler, S.A. Rehner and Humber, comb. nov. [12]		
*Metarhizium brunneum* Petch [25,27]		
*Metarhizium campsosterni* (W.M. Zhang and T.H. Li) Kepler, S.A. Rehner and Humber, comb. nov. [12]		
*Metarhizium carneum* (Duché and R. Heim) Kepler, S.A. Rehner and Humber, comb. nov. [12]		
*Metarhizium cylindrosporum* Q.T. Chen and H.L. Guo [12]		
*Metarhizium dendrolimatilis* Z.Q. Liang, W.H. Chen, Y.F. Han and D.C. Jin, sp. nov. [48]		
*M. flavoviride* (Gams and Rozsypal) [25] x		X
*Metarhizium flavoviride* var. *flavoviride* [29] x		X
*Metarhizium frigidum* J. Bisch. et S. A. Rehner, sp. nov. [40] x		X
*Metarhizium globosum* J.F. Bisch., Rehner and Humber sp. nov. [25,27]		
*Metarhizium granulomatis* (Sigler) Kepler, S.A. Rehner and Humber, comb. nov. [12]		
*Metarhizium guizhouense* Q.T. Chen and H.L. Guo, anamorph of M. taii [25,27]	*Metarhizium taii* Z.Q. Liang and A. Y. Liu	
*Metarhizium guniujiangense* (C.R. Li, B. Huang, M.Z. Fan and Z.Z. Li) Kepler, S.A. Rehner and Humber, comb. nov. [12]		
*Metarhizium indigoticum* (Kobayasi and Shimizu) Kepler, S.A. Rehner and Humber, comb. nov. [12]		
*Metarhizium khaoyaiense* (Hywel-Jones) Kepler, S.A. Rehner and Humber, comb. nov. [12]		
*Metarhizium koreanum* Kepler, S.A. Rehner and Humber, sp. nov. [12] x		X
*Metarhizium kusanagiense* (Kobayasi and Shimizu) Kepler, S.A. Rehner and Humber, comb. nov. [12]		
*Metarhizium lepidiotae* [27] (Driver and Milner) J.F. Bisch., Rehner and Humber stat. nov. [25,27]	*Metarhizium anisopliae* var. *lepidiotae* Driver and Milner (as *Metarhizium anisopliae* var. *lepidiotum*) [29]	
*Metarhizium majus* (J.R. Johnst.) J.F. Bisch., Rehner and Humber stat. nov. [25,27]	*Metarhizium anisopliae* var. *major* (J.R. Johnst.) M.C. Tulloch [46]	
*Metarhizium marquandii* (Massee) Kepler, S.A. Rehner and Humber, comb. nov. [12]		
*Metarhizium martiale* (Speg.) Kepler, S.A. Rehner and Humber, comb. nov. [12]		
*Metarhizium minus* (Rombach, Humber and D.W. Roberts) Kepler, S.A. Rehner and Humber, comb. et stat. nov. [12] x	*Metarhizium flavoviride* var. *minus* Rombach, Humber and D.W. Roberts [29] x	X
*Metarhizium novozealandicum* Kepler, S.A. Rehner and Humber, comb. et stat. nov. [12]	*Metarhizium flavoviride* var. *novozealandicum* Driver and R.J. Milner [29]	
*Metarhizium owariense* (Kobayasi) Kepler, S.A. Rehner and Humber, comb. nov. [12]		
*Metarhizium owariense* f. *viridescens* (Uchiy. and Udagawa) Kepler, S.A. Rehner and Humber, comb. nov [12]		
*Metarhizium pemphigi* (Driver and R.J. Milner) Kepler, S.A. Rehner and Humber, comb. et stat. nov. [12] x	*Metarhizium flavoviride* var. *pemphigi* Driver and R.J. Milner [29] x	X
*Metarhizium pingshaense* Q.T. Chen and H.L. Guo [25,27]		
*Metarhizium purpureogenum* O. Nishi, S. Shimizu, H. Sato, sp. nov. [14] x		X
*Metarhizium pseudoatrovirens* (Kobayasi and Shimizu) Kepler, S.A. Rehner and Humber, comb. nov. [12]		
*Metarhizium rileyi* (Farl.) Kepler, S.A. Rehner and Humber, comb. nov. [12]		
*Metarhizium robertsii* J.F. Bisch., Rehner and Humber sp. nov. [25,27]		
*Metarhizium taii* Z.Q. Liang and A.Y. Liu [12]		
*Metarhizium yongmunense* (G.H. Sung, J.M. Sung and Spatafora) Kepler, S.A. Rehner and Humber, comb. nov. [12]		
*Metarhizium viride* (Segretain, Fromentin, Destombes, Brygoo and Dodin ex Samson) Kepler, S.A. Rehner and Humber, comb. nov. [12]		
*Metarhizium viridulum* (Tzean, L.S. Hsieh, J.L. Chen and W.J. Wu) B. Huang and Z.Z. Li [12]		

**Table 2 insects-10-00385-t002:** The effect of fermentation, formulation and the efficacy of *Metarhizium flavoviride* sensu lato under greenhouse and open field conditions. We selected studies that provide suggestions for the application of *M. flavoviride* s. lato against arthropod pests. Note that before the general spread of rDNA sequence data-based examination (i.e., before 2000) species identification used to rely on conidial shape and size, so earlier publications using acridid pests as target species may either actually have *M. flavoviride* s. lato or *M. acridum* as their fungal agent. Isolate FI985 for example should be named *M. acridum*.

	**Effect of Fermentation on the Performance of *Metarhizium flavoviride***
**Year**	**Culture Variable**	**Other Conditions**	**Measured Outcome**	**References**	**Observations**
1993	Medium content	na.	Conidial production	Jenkins and Prior 1993 [65]	First record of *M. flavoviride* in submerged culture
1993	Medium content	Conidial concentration	Mortality of targets and secondary sporulation	Lomer et al., 1993 [50]	
1994	Medium content, incubation temperature	Length of storage, high temperature after storage	Germination rate	McClatchie et al., 1994 [66]	
1996	Age of culture	Stored as powder or oil-formation, silica gel	Viability, moisture content	Moore et al., 1996 [67]	
1997	Addition of fatty acids of various chain lengths	Acids added, time, previous storage; inhibitor and promoter	Germination percentage	Barnes and Moore 1997 [68]	
1997	C:N ratio, age of culture (time passed after inoculation)	Air-drying, temperature and method (dry powder or in oil)	Conidial viability (germination)	Moore and Higgins 1997 [69]	
2000	Length of incubation after inoculation	Speed of drying	Germination rate	Hong et al. 2000 [96]	
2001	Growing media	Lighting regime	Diameter of colonies, number of conidia	Onofre et al. 2001 [97]	
2001	Growing media	na.	Conidial production, morphological features, fungal pathogenicity	Fargues et al. 2002 [70]	
2005	N sources in the medium; C:N ratio; amount of oxygen	pH regulation	Conidial growth, production of conidia and blastospores	Issaly et al. 2005 [71]	First record of culture parameters on blastospores in submerged culture
	**Effect of formulation on the performance of *Metarhizium flavoviride***
	**Experimental condition or variable**	**Type of formulation**	**Reference**	**Note**
1993	Wavelength of solar radiation, age of cultures, oils added, sunscreens	Sunscreen compounds dissolved in oils	Moore et al., 1993 [73]	Simulated UV-exposure in a laboratory
1993	Formulation, relative humidity, conidial concentrations	Oil- or water-based formulation	Bateman et al., 1993 [74]	Efficacy against target organism was tested
1993	Length and temperature of storage, oils, drying	Oils of mineral, vegetable and animal origin, molasses	Stathers et al., 1993 [98]	A storage experiment
1994	Formulation type	Oil-based and water-based formulations	Ball et al., 1994 [75]	Formulation types and dosages on targets and non-targets
1994	Length of sunlight, time	Sunscreens	Hunt et al., 1994 [76]	
1994	Oil type, silica gel, temperature, time	Vegetable oils mixed with mineral oils	McClatchie et al., 1994 [66]	
1995	Storage time and temperature, addition of antioxidants, silica gel	Vegetable or mineral oils	Moore et al., 1995 [86]	Efficacy against target organism
1995	Oil type and degree of refinement, time	Vegetable and mineral oils	Prior et al., 1995 [84]	Stand-alone toxicity of oils that are optional in formulations
1996	Oils, silica gel; storage with or without formulation; storage time and temperature	Addition of oils to conidia	Moore et al., 1996 [67]	
1996	Type and freshness of bait	Ingredients within bait	Caudwell and Gatehouse 1996 [85]	
1996	Formulation type, temperatures, incubation temperature, storage time	Oil and dry formulation, silica gel	Morley-Davies et al., 1996 [99]	
1997	Clay types, storage temperature, an oil mix	Minerals to conidial suspension, a mineral oil-mixture	Moore and Higgins 1997 [69]	
1997	Types of clay, storage temperature, addition of an oil mix	Minerals to conidial suspension, a mineral oil-mixture	Moore and Higgins 1997 [69]	Types of clays, surface areas
1997	Temperature, inoculation method, spore carrier, relative humidity	Oil suspensions and aqueous suspension	Ouedraogo et al., 1997 [88]	Efficacy against target organisms. Carrier type and inoculation method
1998	Sunscreen oil, time after treatment, time of application	Oil suspensions	Shah et al., 1998 [100]	Caged field trial. Efficacy against target organisms
2015	Nano technique, conidial concentrations	Nano-formulated fungus	Sabbour 2015 [77,78]	Efficacy against target organisms
2015	Solar radiation, time, oils and sunscreens.	Oil suspension	Fernandes et al. 2015 [101]	UV-tolerance and the country of origin
	**Laboratory studies and caged field trials testing the effect of *M. flavoviride***
	**Conditions**	**Targeted Order: Family**	**Method**	**Reference**	**Note**
1983	Temperature, conidial concentrations	Coleoptera: Curculionidae	Spraying with a spray tower apparatus	Soares et al., 1983 [102]	
1987	None	Homoptera: Delphacidae	Conidial suspension by spinning disc applicator	Aguda et al., 1987 [79]	Caged field trial. *M. flavoviride* var. *minus*
1992	Conidial concentration	Orthoptera: Acrididae	Fungal inoculum applied to the body of targets	Moore et al., 1992 [80]	
1993	Formulation type, conidial concentration, relative humidity	Orthoptera: Acrididae	Topical administration	Bateman et al., 1993; Lomer et al., 1997 [51,74]	
1993	Various isolates	Coleoptera: Curculionidae	Immersion to conidial suspension	Moorhouse et al., 1993 [81]	*M. flavoviride* var. *minus*
1993	Conidial concentrations	Orthoptera: Acrididae and Phalacridae	Inoculated at the mouthpart of the body	Milner and Prior 1994 [82]	
1994	None	Orthoptera: Acrididae	Conidial suspension applied topically	Seyoum et al., 1994 [83]	Flight and feeding behaviour
1994	Temperature	Orthoptera: Acrididae	Spraying with blastophore suspension	Welling et al., 1994 [34]	
1995	Age and sex of targets, fungal concentration and site of inoculation, various formulation oils	Orthoptera: Acrididae	Inoculation with conidial suspension	Prior et al., 1995 [84]	
1995	Formulation oils, presence of an antioxidant, humidity of product, storage conditions	Orthoptera: Acrididae	Topical administration	Moore et al., 1995 [86]	
1996	Conidial concentrations, components of bait	Orthoptera: Acrididae	Feeding bait inoculated with fungus	Caudwell and Gatehouse 1996 [85]	
1996	Freshness of bait	Orthoptera: Acrididae	Baited feeding	Caudwell and Gatehouse 1996 [85]	Caged field study
1996	Conidial concentrations	Orthoptera: Acrididae	Topical administration	Milner et al., 1996 [87]	
1997	Droplet size, per hectare volume, type of enclosure	Orthoptera: Acrididae	Aerial spray	Price et al., 1997 [103]	Caged field and enclosed field study
1997	Fungal isolates, temperature	Orthoptera: Acrididae	Inoculated at body parts	Milner 1997 [36]	
1997	Basking, temperature, combination with another fungus.	Orthoptera: Acrididae	Inoculated feed	Inglis et al., 1997 [104]	
1997	Method of fungal administration, temperature and humidity	Orthoptera: Acrididae	Topical inoculation or spray	Ouedraogo et al., 1997 [88]	
1997	Conidial concentration, method of fungal administration	Coleoptera: Coccinellidae and Tenebrionidae, Neuroptera: Myrmeleontidae, Araneae: Philodromidae, Orthoptera: Acrididae	Exposure to leaves treated with fungal suspension, fungus-treated feed, topical administration	Peveling and Demba 1997 [89]	Study aimed at non-target arthropods.
1997	Temperature and conidial concentration	Orthoptera: Pyrgomorphidae	Topical administration	Thomas and Jenkins 1997 [105]	
1997	Spore concentrations	Orthoptera: Pyrgomorphidae	Topical administration	Thomas et al., 1997 [106]	Caged field study
1997	Temperature, *Beauveria bassiana* (Balsamo-Crivelli) Vuillemin (Hypocreales: Cordycipitaceae)	Orthoptera: Acrididae	Feeding inoculated leaves	Inglis et al., 1997 [104]	
1997	Moisture content of dehydrated conidia, rehydration time	Orthoptera: Acrididae	spray with a medium-droplet applicator	Moore et al., 1997 [107]	Caged field study
1998	Temperature and relative humidity, various life stages of pest	Orthoptera: Acrididae	Ultra-low volume-spray	Sieglaff et al., 1998 [108]	Caged greenhouse, a follow-up study
1998	Conidial concentrations, age of fungal cultures	Orthoptera: Acrididae	Spray or topical administration	Sieglaff et al., 1998 [108]	This was the initial study
1998	Time after treatment, sunscreens	Orthoptera: Acrididae	Spray	Shah et al., 1998 [100]	Caged field study
1999	Conidial concentrations, addition of *B. bassiana*, temperature regimes	Orthoptera: Acrididae	Inoculated feed	Inglis et al., 1999 [109]	
2001	Components of medium	Orthoptera: Acrididae	Leaves treated with fungus	Fargues et al. 2002 [70]	
2008	Isolates, conidial concentrations	Homoptera: Delphacidae	Spore suspension spray	Jin et al. 2008 [110]	With* M. flavoviride* var. *minus*
2011	None	Hemiptera: Reduviidae	Conidial spray	Rocha and Luz 2011 [111]	The first report of *M. flavoviride* var. *pemphigi* against *Triatoma infestans*
2011	Conidial concentrations	Lepidoptera: Noctuidae	Surface contamination	Belen et al. 2011 [57]	
2012	Isolates, life stage of pests	Homoptera: Delphacidae	Fungal suspension spray	Li et al. 2012 [112]	
2014	Presence of a major accumulation pheromone	Orthoptera: Acrididae	Topical application	Gorashi 2014 [113]	Feeding and movements were recorded
2015	Conidial concentrations	Lepidoptera: Noctuidae, Pyralidae and Crambidae	na.	Sabbour 2015 [78]	Nano-formulated fungus
2015	None	Hemiptera: Aphididae	Hand spray or tower spray	Lee et al. 2015 [114]	
2015	Other fungal entomopathogens, method of infection, conidial concentration	Coleoptera: Tenebrionidae	Seed treatment, inoculation, fungus-treated feed	Rangel et al. 2015 [90]	Fungal combinations tested against *Fusarium culmorum*
2015	Conidial concentrations	Lepidoptera: Noctuidae, Pyralidae and Crambidae	Fungus-treated leaves	Sabbour 2015 [77]	
2015	Conidial concentrations	Lepidoptera: Gelechiidae	Leaves treated with fungus	Sabbour 2015 [77]	Nano-formulated fungus
2015	Different strains	Lepidoptera: Pyralidae, Coleoptera: Chrysomelidae, Tenebrionidae and Curculionidae	Topical administration	Kocaçevik et al. 2015 [62]	The fungal isolate used was initially found on larvae of *Amphimallon solstitiale*
2017	Rice variety and temperature, a symbiotic bacterium	Homoptera: Delphacidae	Exposure to fungal suspension	Huanhuan et al. 2017 [115]	
2017	Combinations of temperature and relative humidity	Coleoptera: Chrysomelidae	Conidial suspension	Kryukov et al. 2017 [116]	*M. pemphigi*
2017	Conidial concentrations, spray cover, life stages	Trombidiformes: Tetranychidae	Spraying spore suspension	Dogan et al. 2017 [117]	Petri dish and pot experiments
2018	None	Hemiptera: Alydidae, Lepidoptera: Plutellidae and Coleoptera: Tenebrionidae	Fungal cultures	Kim et al. 2018 [8]	Fungal isolates from a fungal library
	**Effect of* M. flavoviride* under greenhouse and open field conditions**
	**Location**	**Target**	**Crop**	**Method**	**Source**	**Note**
1993	Field	Orthoptera: Pyrgomorphidae	none or mixed vegetables	Spray	Lomer et al., 1993, 1997 [50,51]	The first outdoor trials with *M. flavoviride*
1995	Field	Orthoptera: Pyrgomorphidae	cassava, shrub, chili and other vegetables	Spinning disc sprayer	Douro-Kpindou et al., 1995; Lomer et al., 1997 [51,118]	
1997	Field	Orthoptera: Acrididae and Gryllidae	none	Spray bands, aerial and mounted spray	Lomer et al., 1997; Milner 1997 [36,51]	
1997	Field	Orthoptera: Acrididae	none	Hand-held sprayer	Langewald et al., 1997; Lomer et al., 1997 [51,91]	
1997	Field	Orthoptera	various	Various	Lomer et al., 1997 [51]	A review including caged field studies and formulation studies
2015	Field	Lepidoptera: Crambidae	corn	Spray	Sabbour 2015 [77]	
2015	Field	Lepidoptera: Noctuidae, Pyralidae and Crambidae	corn	Spray	Sabbour 2015 [78]	Nano-formulated fungus
2015	Field and greenhouse	Lepidoptera: Gelechiidae	potato	Spray	Sabbour 2015 [77]	Nano-formulated fungus
	**Studies on the compatibility of management types and agricultural substances on *M. flavoviride***
	**Variable**	**Type of variable**	**Source**	**Note**
1997	Chemical treatments to soil before isolation of fungus, temperature	Herbicide, fungicide and insecticide	Mietkiewski et al., 1997 [53]	Laboratory test
2011	Concentration of chemical	Fungicide	Damin et al. 2011 [92]	Laboratory test
2011	Fertilizer in double and single amount, organic manure	N, P, K, and organic manure	Jarmul-Pietraszczyk et al. 2011 [94]	Field study
2011	Conventional and organic fertilization, presence of plants	N, P, K; pig slurry, green manure	Meyling et al. 2011 [95]	Field study
2016	Management type	Organic, conventional	Sammaritano et al. 2016; de Castro 2016 [93,119]	Collection of fungal entomopathogens directly from the soil [93] or from soil samples for identification and further use
	**Studies on the effect of *M. flavoviride* on non-target species**
	Non-target organism	Conditions of application	Source	Note
1994	Hymenoptera: Apidae	Oil and water-based formulations, conidial dosage, spray	Ball et al. [75],	Caged study
1997	Coleoptera: Coccinellidae and Tenebrionidae, Neuroptera: Myrmeleontidae, Araneae: Philodromidae	Exposure to leaves treated with fungal suspension, fungus-treated feed, topical administration	Peveling et al. [89]	Conidial concentration, method of fungal administration were also studied
1999	Galliformes: Phasianidae	ingestion of spore-coated feed, ingestion of infected insects	Smits et al. [120]

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
