# Peer review of "Current Knowledge of the Entomopathogenic Fungal Species Metarhizium flavoviride Sensu Lato and Its Potential in Sustainable Pest Control"

_insects, 2019, doi:10.3390/insects10110385_

Round 1

Reviewer 1 Report

General comments

This review article have considerable problems in species identification of Metarhizium spp. referred in the review.

This study aimed to review M. flavoviride, however, most of the Metarhizium strains referred in the manuscript is correctly M. acridum. Other than M. acridum, M. pemphigi, M. minus, M. frigidum and maybe some other species are also referred to as M. flavoviride. Only a few references are on M. flavoviride sensu stricto.

Almost all Metarhizium isolates from locusts morphologically identified as M. flavoviride were confirmed to be M. acridum (also known as M. anisopliae var. acridum) in the current taxonomy based on molecular phylogenetics. It is difficult to identify many species in Metarhizium only by morphological characteristics because of small interspecific morphological variations. Thus, literatures on Metarhizium published before the publication of the latest taxonomy of M. flavoviride and M. acridum in Bischoff et al (2006; 2009) include many misidentifications. It is very difficult to review a particular taxonomic group on the basis of literatures before the latest taxonomy.

M. acridum is a specialist parasite of locust distributing widely in tropical and subtropical area. On the other hand, M. flavoviride have been isolated from more than two orders of insect, suggesting their wide host range, and distributes in Europe and Australia. The two species also have opposite thermal growth preferences. Thus, reviews on M. flavoviride without distinction of morphologically indistinguishable species necessarily confuse readers. The authors should understand the current status of Metarhizium taxonomy before writing review on a particular species in Metarhizium.

The authors wrote that they reviewed on M. flavoviride because this species have not been reviewed despite it has been registered as mycoinsecticides. However, as far as I know M. flavoviride sensu stricto has not been registered as insecticides but M. acridum has.

The authors may be able to revise the manuscript to focus on only M. acridum. However, an excellent review on biology and biocontrol perspective of M. acridum have already published by FAO in 2007. For M. flavoviride sensu stricto, very few studies have been published for this species. Thus, reviewing on this species has less necessity.

Specific comment

L31 I believe that M. brunneum F52 maybe have been more studies and used excessively than M. anisopliae.

L36

M. brunneum and M. robertsii have been used as mycoinsecticides for a long time. Most isolates of M. brunneum and M. robertsii were identified as M. anisopliae until recently due to small morphological differences among them. 

L39 "Metarhizium rileyi~"

riley -> rileyi

Some references should be presented.

Metarhizium rileyi is specialist of lepidopteran larvae. It has not been found to infect Coleoptera.

L41

As far as I know M. flavoviride sensu stricto has not been resistered as insecticides

L52

"Materials and Methods" is usually not used for a headline in a review article.

"Literatures on M. flavoviride" may be better.

Author Response

Answer to Reviewer 1

Thank you for your insights and thoughtful, professional comments on our submitted review. We followed carefully all recommendations and we stick to our original purpose and focus on the M. flavoviride sensu lato as described by Schmelz et al. (2017). Please see our point-by-point answer to your recommendations.

Rev. 1. Language/style suggestions

Authors: Done. We carefully followed and accepted all your grammar/language/style suggestions.

Rev. 1. "Materials and Methods"… not used in a review … "Literatures on M. flavoviride" ...

Authors: The headline was changed. Thank you for pointing out this style error for us.

Rev. 1. The authors …reviewed on M. flavoviride because this species have not been reviewed despite it has been registered as mycoinsecticides. … M. flavoviride sensu stricto has not been registered as insecticides but M. acridum has.

Authors: Indeed, M. anisopliae var. acridum has been a registered product. For example in their review Maina et al. (2018) refers to earlier accounts of a registered product that contains M. anisopliae var. acridum. We added the paper by Maina et al (2018) to support the claim that at one point of time, M. flavoviride was considered the main agent of a commercial product. When presenting details on the status of M. flavoviride as a commercial product, Maina et al. (2018) refer to a Milner (2000) paper that uses the Driver et al. (2000) paper as a reference for explaining the difference between M. species.

Rev. 1. L41 As far as I know M. flavoviride sensu stricto has not been registered as insecticides

Authors: Besides the above-mentioned work of Maina et al. (2018) we added two independent papers to support that M. flavoviride s.l. at one point of time was a commercially available product. These two sources are Milner RJ (2000) and Skinner et al. (2014). Also please find Kim et al. (2018), included in our manuscript. Their paper mentions Biogreen as a product that contains M. flavoviride v. flavoviride. Although these four papers do not reach a consensus on the taxonomic stance of the microbial agent within Biogreen (or BioGreen), we are inclined to accept your opinion that M. flavoviride s.str. has not been registered as a bioinsecticide. Therefore, we added sentences to inform the reader that although there was a M.f.-based product, there is a taxonomic ambiguity of the microbial agent. We thank you for making us more aware of this situation.

Rev. 1. "Metarhizium rileyi~"riley -> rileyi

Authors: The typo was corrected.

Rev. 1. Some references should be presented. Metarhizium rileyi is specialist of lepidopteran larvae. It has not been found to infect Coleoptera.

Authors: Correction accepted. We found two supporting independent sources and understand now that the host range of M. rileyi is exceptionally narrow, with Lepidopterans being almost exclusively the host species for this fungus.

Source 1: Devi P.S.V., Prasad Y.G. (2001) Nomuraea Rileyi — A Potential Mycoinsecticide. In: Upadhyay R.K., Mukerji K.G., Chamola B.P. (eds) Biocontrol Potential and its Exploitation in Sustainable Agriculture. Springer, Boston, MA

Source 2: Fronza, E.; Specht, A.; Heinzen, H.; Barros, N.M. de Metarhizium (Nomuraea) rileyi as biological control agent. Biocontrol Sci. Technol. 2017, 27, 1243–1264.

Fronza et al (2017) states that “The host range of M. rileyi is more restricted […] mostly limited to lepidoptera larvae […], although a few beetles, some hemipterans and representatives of a few other arthropod orders are also vulnerable to this fungus (Humber et al., 2011; Ignoffo, 1981; Matter & Sabbour, 2013).” We referred to Fronza et al. (2017) in the original version of the manuscript. Now we changed the wording to be more accurate, and our manuscript does not wish to elaborate on M. rileyi. Thank you for helping us making our work more to-the-point.

Rev. 1. … M. brunneum F52 …. more studies and used [more] excessively than M. anisopliae.

… M. brunneum and M. robertsii have been used as mycoinsecticides for a long time. Most isolates of M. brunneum and M. robertsii were identified as M. anisopliae until recently due to small morphological differences among them.

Authors: We value your experience and opinion and we agreed to modify our manuscript accordingly. We have changed the lines referring to the potential of M. roberstii and M. brunneum, although we kept the emphasis on M. anisopliae being the most well-known and most researched species of the genus. We hope this modification is acceptable.

Rev. 1. …study aimed to review M. flavoviride, however, most … Metarhizium strains referred in the manuscript is M. acridum. Other than M. acridum, M. pemphigi, M. minus, M. frigidum and maybe some other species are also referred to as M. flavoviride. Only a few references are on M. flavoviride sensu stricto. Almost all Metarhizium isolates from locusts morphologically identified as M. flavoviride were confirmed to be M. acridum (also known as M. anisopliae var. acridum) in the current taxonomy based on molecular phylogenetics. It is difficult to identify many species in Metarhizium only by morphological characteristics because of small interspecific morphological variations. Thus, literatures on Metarhizium published before the publication of the latest taxonomy of M. flavoviride and M. acridum in Bischoff et al (2006; 2009) include many misidentifications. It is very difficult to review a particular taxonomic group on the basis of literatures before the latest taxonomy.

Authors: With due recognition of your expertise, we agreed to take our manuscript into a somewhat new direction. We have acquired copies of the two Bischoff et al. papers, studied them and incorporated them to our manuscript upon your advice. We have included a new section to address the ambiguity of taxonomy. At the same time, as we communicated to the Editorial Board as well our review paper is not a mycological feature. We address entomologists and researchers who are relatively new to the field of organic pest management and/or the use of entomopathogenic fungi and/or the genus Metarhizium. Mycologists are well supplied with methods and means of identification, so perhaps this manuscript may not contain as much new information, may not orientate them as much as it may entomologists, who are focusing on a broader sense of a species. Their target is the species they can use.

Rev. 1. M. acridum is a specialist of locust ….in tropical and subtropical area. On the other hand, M. flavoviride … wide host range, and distributes in Europe and Australia. The two species …. opposite thermal growth preferences. Thus, reviews on M. flavoviride without distinction of morphologically indistinguishable species necessarily confuse readers. The authors should understand the current status of Metarhizium taxonomy before writing review on a particular species in Metarhizium.

Authors: Your comment is appreciated. We agreed to change the manuscript to conform to your claims.

Rev. 1. … revise the manuscript to focus on only M. acridum. However, an excellent review on biology and biocontrol perspective of M. acridum …by FAO in 2007. For M. flavoviride sensu stricto, very few studies have been published for this species. Thus, reviewing on this species has less necessity.

Authors: We have acquired a copy of the FAO article written by Harold van der Valk in 2007. It is indeed a thorough compilation and review of studies on the efficacy of M. anisopliae var. acridum. The author argues that he includes all reports referring to M. flavoviride, up to 2007 per se, by pointing out that before a taxonomic revision by Driver et al. (2000), M. anisopliae var. acridum was referred to as M. flavoviride. We have added Driver, F., Milner, R.J., Trueman, J.W.H., 2000. A taxonomic revision of Metarhizium based on a phylogenetic analysis of rDNA sequence data. Mycol. Res. 104, 134–150. https://doi.org/10.1017/S0953756299001756 to our review to make taxonomy issues more precise. Although 12 years have passed since the publication of the 2007 FAO review by Mr. van der Valk, we do not feel the need to write another, a follow-up review on M. anisopliae var. acridum. We wish to keep the focus on any isolate/strain that bears the name M. flavoviride in the literature. Also, we have added a paper by Danfa and van der Valk (1999), to demonstrate problems of taxonomy: “Recent taxonomic studies at molecular level seem to indicate that the isolates listed here as Metarhizium sp. form a homologous group and may be considered as Metarhizium anisopliae var. acridum (Driver et al., in press). A number of these isolates have in the past been referred to as M. flavoviride or M. anisopliae”.

Reviewer 2 Report

As my concern the manuscript is interesting , well described and the major subject is to compile the stdudies and articles published on Metarhizium flavoviridae, which is well organized. only I would suggest to also include citations and references from SouthAmerica and America in general which is mostly not considered in majority of references except some from Brazil. 

Author Response

Answer to Reviewer 2

Thank you for taking your time to read and analyse our work. We are also thankful for your kind words of appreciation. Understanding your concern about the need to include research accounts of the species from the American continent we included a few references. To our knowledge, for South America, that is, most of the research has been done in Brazil, so these few references are apparently the ones that deal with M. flavoviride in regions outside Brazil. With regard to North America, climatic circumstances may be accountable for this phenomenon, but we were surprised at the lack of Central American studies on the species.

Rev. 2. include citations and references from South America and America in general … except … from Brazil.

Authors: Done, we do all our best to add these works but only very few were found.

Fisher, J.J.; Rehner, S.A.; Bruck, D.J. Diversity of rhizosphere associated entomopathogenic fungi of perennial herbs, shrubs and coniferous trees. J. Invertebr. Pathol. 2011, 106, 289–295. Done.

This paper investigated the fungal communities of the rhizosphere of various crops located in USA, North America.

Smits, J. E., Johnson, D. L., & Lomer, C. (1999). Pathological and Physiological Responses of Ring-Necked Pheasant Chicks Following Dietary Exposure to the Fungus Metarhizium Flavoviride, a Biocontrol Agent for Locusts in Africa. Journal of Wildlife Diseases, 35(2), 194–203. DOI:10.7589/0090-3558-35.2.194. We added this paper to our review. This paper investigates the effect of M. flavoviride on a non-target avian species. The study took place in Canada, North America.

J.A. Aguilera Sammaritano, C.C. López Lastra, A. Leclerque, F. Vazquez, M.E. Toro, C.P. D’Alessandro, A.G.S. Cuthbertson & B.E. Lechner (2016) Control of Bemisia tabaci by entomopathogenic fungi isolated from arid soils in Argentina, Biocontrol Science and Technology, 26:12, 1668-1682, DOI: 10.1080/09583157.2016.1231776. We added this paper to our review. Although not confined to M. flavoviride, the study uses Metarhizium isolates collected with the Galleria bait method in Argentina, South America.

Hernández-Velázquez V.M., Berlanga-Padilla A.M., Garza-González E. 1997. Detección de Metarhizium flavoviride sobre Schistocerca piceifrons piceifrons (Orthoptera: Acrididae) en la Isla Socorro, Archipiélago de Revillagigedo, México. Vedalia 4: 45-46. We added this paper to our review. The study confirms the presence of M. flavoviride in an Orthopteran host in Mexico, Central America.

Reviewer 3 Report

Dear authors,

I've read the manuscript entitled Current knowledge of the entomopathogenic fungal species Metarhizium flavoviride and its potential use in sustainable pest control by Franciska Tóthné Bogdány et al. The authors have performed a rigorous literature survey and study on the topic of an under-researched entomopathogenic fungus (EPF), Metarhizium flavoviride. The study provides a holistic overview of some of the most important fungal traits, including natural occurrence, effects of formulation on its pathogenicity, and its efficacy in lab, glasshouse and field trials, to mention a few of the important topics addressed in this review.

I found the manuscript interesting and concise, with only a few minor language issues. However, I believe this is the paper’s only weak point.

Based on this I suggest accepting this paper with minor revisions as suggested in the attached PDF file.

Best regards,

Anonymus reviewer.

Minor comments:

See comments in the attached PDF file.

Author Response

Answer to Reviewer 3

Thank you for taking your time and pointing out our mistakes. Some of the grammar ones were really basic ones and we feel sorry for those. We hope the new corrected version will get your fullest approval.

Rev. 3. a few minor language issues… minor revisions as suggested in the attached PDF file

Authors: Done. All language suggestions were duly followed, but at some places the text, due to following the requests of other reviewers was changed and the submitted version no longer is there. Suggestions for minor revisions are dealt with below. For more clarity, we used the line numbers of the submitted version.

Rev. 3. Consider mentioning regarding M. brunneum vs. wireworms: RAZINGER, Jaka, SCHROERS, Hans-Josef, UREK, Gregor. Virulence of Metarhizium brunneum to field collected Agriotes spp. wireworms. Journal of agricultural science and technology, 2018, vol. 20, iss. 2, pp. 309-320. http://jast.modares.ac.ir/article-23-9716-en.pdf.

Authors: Done. Indeed, we failed to support the claim in this sentence. We added this reference and another one: Eckard, S.; Ansari, M.A.; Bacher, S.; Butt, T.M.; Enkerli, J.; Grabenweger, G. Virulence of in vivo and in vitro produced conidia of Metarhizium brunneum strains for control of wireworms. Crop Prot. 2014, 64, 137–142.

Rev. 3. Consider mentioning also, regarding M. brunneum and a fly Delia radicum:

HERBST, Malaika, RAZINGER, Jaka, UGRINOVIĆ, Kristina, ŠKOF, Mojca, SCHROERS, Hans-Josef, HOMMES, Martin, POEHLING, Hans-Michael. Evaluation of low risk methods for managing Delia radicum, cabbage root fly, in broccoli production. Crop protection, 2017, vol. 96, 273-280, doi: 10.1016/j.cropro.2017.02.023 and/or

RAZINGER, Jaka, Ĺ˝ERJAV, Metka, URBANÄŚIÄŚ ZEMLJIÄŚ, Meta, MODIC, Špela, LUTZ, Matthias, SCHROERS, Hans-Josef, GRUNDER, Jürg M., FELLOUS, Simon, UREK, Gregor. Comparison of cauliflower-insect-fungus interactions and pesticides for cabbage root fly control. Insect science, 2017, vol. 24, iss. 6, pp. 1057-1064, doi: 10.1111/1744-7917.12534 and/or

RAZINGER, Jaka, LUTZ, Matthias, SCHROERS, Hans-Josef, PALMISANO, Marilena, WOHLER, Christian, UREK, Gregor, GRUNDER, Jürg M. Direct plantlet inoculation with soil or insect-associated fungi may control cabbage root fly maggots. Journal of invertebrate pathology, ISSN 0022-2011, 2014, vol. 120, no. , str. 59-66, doi: 10.1016/j.jip.2014.05.006.

Authors: Done. We incorporated the first one (with Delia radicum as a target pest), but not the other two to our present manuscript, because we would like to focus on the original species (M. flavoviride).

Rev. 3. Different fonts used?

Authors: Done. it must be a technical problem. Before submission, we all checked the way the manuscript looked like, and found it right. Besides, this oftentimes happens to us as well: we see certain lines of a .pdf document appear in a totally different font. It usually happens around the end/beginning of a page.

Rev. 3. L 56. Why we have not used Web of Science

Authors: Done. We are not regular users of Web of Science, but upon your suggestion, we visited the site to find three papers that became important additions to our revised manuscript. These are: Danfa and Valk, 1999., Hernández-Domínguez et al. (2016) and Schmelz et al., 2017. Thank you for reminding us to using Web of Science.

Rev. 3. L. 58-63 A long, somewhat unclear sentence with improper punctuation.

Authors: Done. We edited the sentence.

Round 2

Reviewer 1 Report

I do not think the review on M. flavoviride sensu lato worth publishing. Species composed of M. flavoviride sensu lato have similar morphology but they differ in host range, geographical distribution, thermal growth preferences, phylogenetic positions. Thus, summarization of information on M. flavoviride sensu lato without distinction of the morphologically indistinguishable species necessarily confuse and are less helpful for readers even though complicated taxonomic situations of this group were presented in advance. The list of literatures on bioassays, laboratory and field studies, fermentation and formulation of M. flavoviride sensu lato summarized in Table 1 is also less helpful for readers.

I can summarize this review by two sentences as follows:
1. Many literatures have been published for various aspect of M. flavoviride sensu lato.
2. However, most of the literatures included misidentification of fungal species.
How do readers make good use of this review?

The authors said that they reviewed this taxonomic group because it is less rarely investigated and documented fungal entomopathogens.
However, M. acridum, one species composing M. flavoviride sensu lato, has been well studied and been commercialized as mycoinsecticides. A review on M. acridum for control of locusts were already published. Thus, this review should exclude M. acridum.

When M. acridum is excluded from this review, only small number of literatures are available for M. flavoviride sensu lato.
Thus, this taxonomic group need more researches (original article) rather than reviews. If the authors are interested in this group, the authors should do research on this group rather than review.

Author Response

Itemized answer to Editors

Academic Editor 1: Decision

Accept after minor revision

Notes for Authors: The manuscript has been well revised and is vastly improved from the orginal submission. I particularly like how the authors have responded to previous concerns about the taxonomy. Their approach is practical and well suited to the context of the article. I still feel that the article is more historical than critical, and would have liked to see some more recommendations coming from the aithors' own insights. Nevertheless the article is suitable for publication. I have listed places where the text needs revision. The writing is a little untidy and I would appreciate if the authors could ensure that the paper is proof read by a native English speaker before the final version is submitted.

Authors:

Dear Editor.

We appreciate your effort in improving our manuscript and special thanks for your decision.

We made a detailed check in the manuscript and corrected all your observations. Additionally we asked one native English colleague to have a detailed check on the manuscript.

At the same time, we do appreciate your suggestion to do original research with this taxonomic group, although the aim of our study will not be of taxonomic nature. We are planning to investigate the potential of entomopathogenic fungi, with possibly M. flavoviride included, in enhancing soil microbiome, and soil characteristics in general when applied in combination with composted municipal waste in various vegetables in greenhouses.

Additionally we had to mention that we also checked the reference list and one change has been made in main text (Castro et al., 2016) and in reference list, position 119.

Please find below our point-by-point answer to your observations, all changes are marked accordingly:

Yellow marks former changes after first review round

Track change marks current changes

The following are some minor corrections: 13 gain more = are gaining increasing

Authors: Done

19 collected = compiled

Authors: Done

22 please revise the sentence ‘we provide information …..assessments, this portion is difficult to comprehend

Authors: Done

28 As an alternative ….

Authors: Done

28 are = is

Authors: Done

33 …related scientific research, including several review articles, and in its use in organic ……

Authors: Done

35 To mention some of the other entomopathogens that have been researched, ….

Authors: Done

37 are also on

Authors: Done

41 have strong potential

Authors: Done

45 refer to the agent

Authors: Done

48 remained = remain

Authors: Done

52 status = state

Authors: Done

66 Starting with the basic …… = ; going from basic toward more specialized information regarding ….  

Authors: Done

70 in fields

Authors: Done

72 … and observational studies were compiled. Authors: Done

74 Metarhizium in italics

Authors: Done

79 explanation for the ubiquity

Authors: Done

85 … on their bodies. In the absence of a suitable host, members of the genus, due to their labile …..

Authors: Done

88 Questions of taxonomy and identification issues

Authors: Done

Simple identification measures have been used with some success: An earlier …. [note that a colon is followed by an uppercase letter]

Authors: Done

upon = based on

Authors: Done

97 a resent attempt to simplify (or clarify) the identification ….

Authors: Done

101 remove ‘even more’

Authors: Done

111-112 check font size

Authors: Done

114-115 Not a new paragraph

Authors: Done

116 specimens

Authors: Done

117 … also becomes unreliable.

Authors: Done

135 of 2017 = conducted in 2017

Authors: Done

135 describing new species

Authors: Done

Table 1 This is an important table and therefore important to get right. It is currently confusing because some of the rows are shaded and others are not. This may give an impression that the non-shaded rows are less important. Perhaps the table could be presented as a simple list with the ‘former name’ indicated in parentheses after its synonym.

Authors: Thank you for this observation. We tried to use these information as simple list, but if we do this, it will be a very long scientific name list in meddle of the main text. As all references are important from this table, we eliminated grey colorations. Indeed all references and information are important from this table. Hope this form is suitable and do not disturb readers.

148 were found

Authors: Done

152 had been made

Authors: Done

152-153 Not a new paragraph. ‘However, the following ….trend.’

Authors: Done

155 a closer relationship

Authors: Done

160 explain MSFC

Authors: Done

160-161 Not a new paragraph

Authors: Done

161 An isolate …

Authors: Done

174 … another continent, Australia, from a native …….in 1979 [36].

Authors: Done

175 from the early 1990’s

Authors: Done

176 was detected on heavily infected …..

Authors: Done

178 remove ‘in 1991’ this is already included in ‘early 1990’s’

Authors: Done

183 ..orthopterans, Ornithacris …..

Authors: Done

185 …in cultivated fields ….

Authors: Done

189 Orthopterans = orthopterans

Authors: Done

194 resolved = concluded

Authors: Done

190-194 check font size

Authors: Done

199 …Slovakia [55], ….

Authors: Done

202 …in the field….

Authors: Done

202 remove ‘if at all’

Authors: Done

206 …as the genus

Authors: Done

221 … that had been set aside ….

Authors: Done

224 Coleopteran = coleopteran

Authors: Done

231 …revealed high diversity within the species …

Authors: Done

236 investigated = investigating

Authors: Done

239 … but this does not imply ….isolated from all soil samples ….

Authors: Done

241 ….that in undisturbed …. Natural vegetation is …..

Authors: Done

245 whether yields are similar between lands under permaculture or conventional ….

Authors: Done

247 As researchers continue to seek biological ….

Authors: Done

250 …..to its more frequent, and possibly more successful use in biological control.

Authors: Done

252 Explain ‘fermentation’

Authors: Done

255 According to the results, the cumulative …..

Authors: Done

257 …. As an oil formulated product.

Authors: Done

258 effect = effects

Authors: Done

260 …carbon:nitrogen [C:N] ratio …. i.e., abbreviation should be explained on first mention

Authors: Done

267 ….past two decades. Therefore, the impact …..

Authors: Done

271 …from the laboratory to the greenhouse ….

Authors: Done

273 ..arable fields, mainly depends …

Authors: Done

by = through

Authors: Done

mineral, natural oil and/or water …

Authors: Done

under various conditions including different wavelengths of …..

Authors: Done

282 …been conducted since that time.

Authors: Done

286 ….such as Ostrinia …..

Authors: Done

287 remove ‘however,’

Authors: Done

290-292 – please revise this sentence

Authors: Done

294 were = have been

Authors: Done

were = was

Authors: Done

body was used against Orthoptera

Authors: Done

revise ‘innoculated at the mouthpart of the body’ do you mean ‘inoculation near the mouthparts’

Authors: Done

306 inoculation = inoculated

Authors: Done

307 administered with the fungus

Authors: Done

309 to clarify is this also related to reference [89]?

Authors: Done

311 …Crambidae; when ….

Authors: Done

312 …offered ad libitum as …. Tenebrionidae; and where the administration …..

Authors: Done

314 ..the number and stable frequency of bioassay studies over the decades.

Authors: Done

315 faces new challenges associated with the ….unknown, M. flavoviride may offer a solution and screening for its efficacy against invasive arthropods is advisable.

Authors: Done

324 …the effects of the fungus ….. in a mixed-vegetable field.

Authors: Done

335 …variable in the agricultural environment …. The use of chemical and organic pesticides and fertilizers.

Authors: Done

342 …or the co-presence ….

Authors: Done

344-348 This summary of the experiments is very difficult to follow. Can the authors please clarify the text.

Authors: Done

349 awaits further studies ….

Authors: Done

…in organic and conventional ….

Authors: Done

remove ‘Metarhizium was found in both locations.’

Authors: Done

353 involved in a small …

Authors: Done

361 scientific attention.

Authors: Done

363 also yet to …… [remove as well]; effect = effects

Authors: Done

364-364 will be put = should be put x2

Authors: Done

373 effect = effects

Authors: Done

374 effect = effects

Authors: Done

377 ….that provide suggestions for the application of

Authors: Done

377 Remove ‘Please’

Authors: Done

Academic Editor 2: Decision:Accept after minor revision

Notes for Authors

I am in agreement with the editors decision --accept with minor revisions as outlined by the editor.

Authors: Thank you for your decision. Please find out itemized answer and changes made in main Ms. Text.

Altogether we highly appreciate the correct and fast process.

With best wishes

Prof. Adalbert Balog

In behalf of all authors

Dated: 28 October, 2019